# Role of Secretory Mucins in the Occurrence and Development of Cholelithiasis

**DOI:** 10.3390/biom14060676

**Published:** 2024-06-10

**Authors:** Zeying Zhao, Ye Yang, Shuodong Wu, Dianbo Yao

**Affiliations:** Department of General Surgery, Shengjing Hospital of China Medical University, No. 36, San Hao Street, Heping District, Shenyang 110004, China; 2022121290@cmu.edu.cn (Z.Z.); 2021121205@cmu.edu.cn (Y.Y.); 2022121321@cmu.edu.cn (S.W.)

**Keywords:** cholelithiasis, mucin, cholesterol stones, pigment stones

## Abstract

Cholelithiasis is a common biliary tract disease. However, the exact mechanism underlying gallstone formation remains unclear. Mucin plays a vital role in the nuclear formation and growth of cholesterol and pigment stones. Excessive mucin secretion can result in cholestasis and decreased gallbladder activity, further facilitating stone formation and growth. Moreover, gallstones may result in inflammation and the secretion of inflammatory factors, which can further increase mucin expression and secretion to promote the growth of gallstones. This review systematically summarises and analyses the role of mucins in gallstone occurrence and development and its related mechanisms to explore new ideas for interventions in stone formation or recurrence.

## 1. Introduction

Cholelithiasis is a prevalent hepatobiliary disease that affects approximately 10–15% of the global population [1,2]. In the United States, this can result in a direct cost of over 6.3 billion dollars per year [3], which poses a significant financial burden. With improvements in living conditions and changes in dietary habits, especially in developing countries, the incidence of gallstones is gradually increasing [4,5,6], and the related economic burden is increasing [7,8,9]. However, the exact mechanism underlying gallstone formation remains unclear, which greatly hampers its prevention and treatment. Gallstone formation is a complex process involving various factors. This review aimed to systematically summarise and analyse the roles of mucin in gallstone development and progression and related mechanisms to provide new insights for intervention in the formation or recurrence of stones (Table 1).

## 2. Literature Search

The literature for this review was selected through a search of PubMed to identify relevant English papers published between 1 January 1980 and 31 March 2024. The search terms ‘Mucin’, ‘MUC2′, or ‘MUC5AC’ were used in combination with the ‘AND’ operator for the terms ‘cholelithiasis’, ‘gallstone’, ‘cholecystolithiasis’, ‘choledocholithiasis’, or ‘hepatolithiasis’. The final reference list was generated based on originality and relevance to the broad scope of this review, with an emphasis on recently published papers.

## 3. Mucin Is Involved in Gallstone Formation

Mucins, a family of large, complex, glycosylated proteins, are secreted by the biliary duct and gallbladder epithelium in the biliary system [10]. The core of mucins typically consists of two domains. One of the domains, an important mucin structure, is the PTS domain with long tandem repeat units containing proline, threonine, and serine, which are sites of heavy O-glycosylation by O-linked oligosaccharides with strong resistance to proteases [11,12,13]. The other domain is non-glycosylated, rich in serine, glycine, glutamine, and glutamic acid, and sensitive to proteases [14,15]. Based on their structural and functional features, mucins can be classified into transmembrane and secretory gel-forming mucins. Transmembrane mucins such as MUC1 and MUC13 are attached to the cell membrane and cover the apical surface. In contrast, secretory gel-forming mucins such as MUC2, MUC6, MUC5AC, and MUC5B are secreted [16]. Mucins, especially secretory mucins, are present in the bile. Mucins also exist in cholesterol and pigment stones [17], and their roles in the pathogenesis of gallstones have been gradually recognised [18,19].

Based on their composition, gallstones are classified as cholesterol, pigment, or mixed stones. Cholesterol stones are more common in Western countries, accounting for approximately 85% of all cases. In contrast, pigment stones are more prevalent in Asia [20]. Animal studies, including those using diet-induced pigment or cholesterol gallstone models, have revealed that mucin secretion is significantly increased in the biliary system, suggesting a possible role of mucin in gallstone formation. Importantly, in response to a lithogenic diet, an increase in biliary mucin was revealed to occur before the formation of stones [21], indicating that stone formation should follow an increase in mucin concentration, and mucin may contribute to the formation of stones. Mucin plays a vital role in the nuclear formation and growth of cholesterol and pigment stones. Electron microscopy and chemical detection have revealed that cholesterol stones are composed of cholesterol crystals along with lesser quantities of pigments and calcium salts. These components are arranged on a matrix consisting of mucin and proteins [22]. Furthermore, mucins accelerate the fusion and aggregation of cholesterol-enriched unilamellar vesicles [23,24]. Pigment stones, which are primarily composed of mucins, bile salts, cholesterol, fatty acids, and bacterial residues [25], are mainly formed by mucin crystals and exfoliated epithelium [26,27]. These findings suggest that mucin acts as the main pronucleating agent in the formation of cholesterol and pigment gallstones. After stone nucleation, enhanced mucin secretion continues, contributing to the growth of gallstones [28]. The mucin within the stones extends radially or in layers from the central region [29], suggesting mucin’s role in stone growth. Therefore, it is currently suggested that mucin, probably mainly secretory mucin, plays important roles in the formation of cholesterol and pigment stones, including cholesterol crystallisation, the nuclear formation of pigment stones, and the deposition and growth of gallstones [30,31,32,33]. For cholesterol and pigment stones, it was revealed that more mucins are contained in gallbladders with brown pigment stones than those with cholesterol stones or without stones, suggesting that mucin should play a more critical role in the formation of brown pigment stones than cholesterol stones [23,34]. Experimental studies on regulating mucin expression may help control stone formation [35].

## 4. Mucin Gel Promotes Stone Formation

Secretory mucin mainly plays a physiological role by polymerising to form protein gels, such as the mucus barrier on the surface of the gastrointestinal mucosa to protect epithelial cells [36]. Similarly, the physical gelation of mucins in the gallbladder or bile duct can occur through a mechanism similar to multiblock copolymer synthesis, which may contribute to gallstone formation (Figure 1a) [37]. The main secretory mucins that participate in gallstone formation, MUC5AC and MUC2, can play their roles in the formation of a highly viscous gel and mucin network through dimerisation and oligomerisation mediated mainly by the hypoglycosylated cysteine-rich domain [38,39,40,41,42]. The N- and C-terminal von Willebrand factor type D and C domains (vWD/vWC) can also be modified by forming disulfide bonds to participate in dimer and multimer formation [43,44,45]. In solution, these polymers are noncovalently linked through electrostatic and hydrophobic interactions, hydrogen bonds, and weak van der Waals forces, leading to gel formation [37].

Mucin gels can be considered the precursor of stones to promote stone formation [46,47]. Typically, mucin gel initially accumulates in the glandular crypts of the gallbladder mucosa, and subsequent crescent accumulation facilitates accelerated nucleation [48]. In addition, the non-glycosylated mucin domain contains hydrophobic binding sites that can bind to substances such as cholesterol, phospholipids, and bilirubin, and this characteristic can also create favourable conditions for stone nucleation [29]. The binding of the hydrophobic domain to bilirubin and calcium in bile leads to the formation of a water-insoluble complex of mucin glycoprotein and calcium bilirubin. In addition, mucin gel can contribute to gallbladder stone formation by facilitating the nucleation of cholesterol monohydrate crystals in the bile, which is already supersaturated [49]. The mucin glycoprotein and calcium bilirubin complex can provide a surface for the nucleation of cholesterol hydrate crystals and serve as a matrix framework that promotes the growth of stones [50]. Specific disruption of the non-glycosylated domain that binds 1-anilino-8-naphthalenesulfonic acid results in a significantly reduced incidence of crystal nucleation stimulated by mucin in model bile, further suggesting that the hydrophobic domain plays a critical role in the nucleation of cholesterol monohydrate crystals [51]. Mucin gel, which serves as a matrix and scaffold for stone growth, promotes stone formation and growth (Figure 2a).

## 5. The Interactions between Mucin and Vesicles Affect Cholesterol Stone Formation

The gallbladder, a part of the extrahepatic biliary tract, is the main site of cholesterol stone formation, mainly because it is the site where bile is stored and concentrated. Bile is composed mainly of water, electrolytes, cholesterol, bilirubin, bile salts, phospholipids, and mucin. It is essential for the digestion of fats and excretion of most cholesterol in the body [52]. After bile is concentrated in the gallbladder by the reabsorption of water and electrolytes, cholesterol, mucin, and bilirubin concentrations in the gallbladder are higher. However, under most conditions, the bile components are in a state of dynamic equilibrium. Cholesterol, which is insoluble in water, is primarily transported in bile through two carriers: bile salt-rich micelles and phospholipid-rich vesicles [53]. When bile is supersaturated with cholesterol, the vesicles become thermodynamically unstable and tend to expand and transform into other structures. Monolayer vesicles in mucin gels can fuse to form multilayer vesicles and liquid crystals, as visualised by low-power polarising microscopy [54]. Liquid crystals or large liposomes resulting from the aggregation and fusion of vesicles can be transformed into cholesterol crystals [55,56]. Cholesterol monohydrate crystals play a significant role in cholesterol stone formation [57]. These solid crystals can grow into microstones with the help of a mucin gel and eventually into larger macroliths (Figure 2b) [58]. The rapid precipitation of cholesterol crystals in the gallbladder is considered the initial step in cholesterol stone formation [59].

Afdhal et al. quantified the effect of cholecystic mucin on vesicle aggregation and fusion using three physicochemical techniques: transmission electron microscopy, dynamic light scattering, and fluorescence biochemical assays. Without mucin, fusion was slow and took up to 24 h; however, with a physiological concentration of mucin, complete fusion of the vesicles occurred within 6 h [60]. A potential mechanism for vesicle fusion, in which the interaction between vesicles and mucin can lead to several physical and chemical alterations in the vesicle structure, was proposed (Figure 1b). Paul et al. added 2 and 4 mg/mL of purified human gallbladder mucin to model bile and found that, after 10 d, the number of cholesterol crystals in the 4 mg/mL group was significantly higher than that in the other group [61]. When cholesterol crystals were incubated in the model bile containing bovine gallbladder mucin, it showed that bovine gallbladder mucin accelerated the growth of cholesterol crystals in supersaturated model bile in a time- and concentration-dependent manner [62,63]. In a study by Wilhelmi et al., adding human or bovine mucin to bile specimens from patients with gallstones yielded similar results [64]. These findings suggest that mucin aids in the nucleation of cholesterol crystals and plays a significant role in their growth and maturation into stones [62].

In addition to mucin, calcium ions have also been identified as a pro-nuclear factor, and their influence on the formation of cholesterol crystals has been investigated [65,66]. Calcium ions and mucin accelerated the nucleation of cholesterol crystals in simulated bile. However, only mucin increased the number of plateau crystals. Mucin’s stimulatory effect was found to depend on the hydrophobic domain. In contrast, calcium ions can increase the binding ability between mucin and bile lipids and pigments [67,68], possibly because mucin is negatively charged and can bind with calcium ions [69,70].

## 6. Bacterial Infections and Inflammation

Bacterial infections of the biliary system are closely associated with cholelithiasis occurrence and progression (Figure 1c) [71,72]. Bacteria in bile and related inflammation are primarily associated with the formation of brown pigment stones [73,74], which are primarily formed in the extrahepatic and intrahepatic bile ducts. The bile is typically sterile. However, bacteria are frequently detected in the bile of patients with cholelithiasis [75,76]. The pathways through which bacteria enter the bile duct include the intestinal barrier and direct reflux from the sphincter of Oddi [77,78]. 16S rRNA gene sequencing technology has shown that the composition of bacterial genera and species in the bile of patients with bile duct pigment stones and the corresponding duodenal intestinal fluid is similar, suggesting that the reflux of bacteria from the duodenum into the common bile duct through the sphincter of Oddi is an important factor in bacterial infection and stone formation in the common bile duct [77,79]. The most common bacteria found in the choledochal bile of patients with cholelithiasis is *Escherichia coli*, followed by *Klebsiella* species [73,77]. Bacterial-derived β-glucuronidase can hydrolyse conjugated bilirubin to unconjugated bilirubin, which combines with calcium ions to produce calcium bilirubinate, the main component of brown pigment stones [80,81]. Moreover, the bacterial glycocalyx, an anionic glycoprotein, can act as a conjugating agent that contributes to the aggregation of calcium bilirubinate crystals, resulting in the formation of stones. The proportion of glycocalyx-producing bacteria in the bile ducts of patients with primary biliary stones is much higher than that of β-glucuronidase-producing bacteria, suggesting that bacterial glycocalyx may have a more significant contribution to the formation of pigment stones than bacterial β-glucuronidase [82].

In addition, accompanied by bacterial infection in the biliary tract caused by intestinal mucosal barrier dysfunction, the level of lipopolysaccharide (LPS), an endotoxin produced by Gram-negative bacteria [83], also increases significantly in the serum. LPS can enter the liver tissue and be secreted into the biliary system, including the intrahepatic and extrahepatic bile ducts and gallbladder [78]. Biliary bacterial infections caused by enterobiliary reflux, such as *Escherichia coli* infection of the biliary tract, can also increase LPS levels in the bile duct [84]. In these cases, LPS can also upregulate the expression of endogenous β-glucuronidase and mucins in bile duct epithelial cells in a dose-dependent manner (Figure 3a). The endogenous β-glucuronidase and mucins can play similar roles with bacterial β-glucuronidase and glycocalyx to promote stone formation [85,86]. Endogenous β-glucuronidase and mucin can play their roles for a longer time, suggesting that the intervention of endogenous β-glucuronidase and mucin expression and secretion might be of greater significance for preventing the formation or recurrence of bile duct pigment stones [87,88].

Inflammation and inflammatory factors are important for developing gallstones (Figure 1d). Inflammation can increase mucin production in the epithelial cells of the gallbladder and bile duct [89]. Vilkin et al. discovered that the gallbladder epithelium exhibited higher levels of MUC5AC when inflamed. This was associated with the formation of pigment stones [90]. Moreover, chronic inflammation of the gallbladder contributes to cholesterol stone formation, and cholesterol monohydrate crystals can induce the secretion of pro-inflammatory cytokines in a T cell-dependent manner [57]. In addition, types of inflammatory factors, such as Tumor necrotic factor-alpha (TNF-α), are responsible for the increased expression of mucin to promote the formation of stones [91,92].

In addition, bacterial infection in the biliary system plays a role in intrahepatic bile ducts and, together with cholestasis, contributes to hepatolithiasis formation [93,94,95]. *Klebsiella* species and *Escherichia coli* are the most common bacteria present in the bile of patients with hepatolithiasis. They are closely related to chronic proliferative cholangitis, promoting the formation of bile duct stones by contributing to mucin production. Pathological examination of patients with proliferative cholangitis revealed chronic inflammation and hyperplasia of the biliary tree and secretory glands, which secrete mucin [83]. Increased remodelling of bile duct epithelial cells resulting from chronic inflammation leads to excessive mucin secretion [18,92], contributing to stone formation. Biliary stones and related infections can further stimulate the proliferation of the biliary tract or even cause biliary strictures, resulting in repeated episodes of chronic proliferative cholangitis, creating a vicious cycle [96].

In clinics, during endoscopic lithotomy, saline irrigation after mechanical lithotripsy has been demonstrated to be effective in reducing the recurrence rate of common bile duct stones, possibly by cleaning stone fragments [97]. In an animal model with chronic proliferative cholangitis, antibiotic irrigation effectively decreased the quantity of residual *Escherichia coli* in the biliary system and reduced the concentration of LPS and the expression of monocyte chemoattractant protein-1 (MCP1), cluster of differentiation 14 (CD14), cyclooxygenase-2, protein kinase C (PKC), Nuclear factor Kappa beta, C-myc, Interleukin(IL)-6, TNF-α, transforming growth factor-β1, β-glucuronidase, and MUC5AC [84]. In addition, LPS can bind to CD14 on bile duct epithelial cells to trigger TNF-α synthesis [94,98], and TNF-α interacts with TNF receptors CD120a and CD120b, leading to PKC activation, which is involved in the upregulation of MUC5AC and MUC2 expression [73,83,94]. LPS can also activate intrahepatic biliary epithelial cells through the Toll-like Receptor 4-Myeloid Differentiation Primary Response Protein 88-dependent pathway, leading to the production of other chemokines or cytokines such as IL-8, MCP-1, and IL-6 [99], which may also participate in the regulation of mucin secretion and contribute to stone formation. Further exploration of the related mechanisms may help identify more effective measures to intervene in the formation of stones instead of simple antibiotic treatment.

## 7. Mucin Hypersecretion Causes Cholestasis and Reduces Gallbladder Activity

In addition to bile constituents, bile flow affects gallstone formation. The hydrodynamic radius of gallbladder mucin is 630A. When the mucin concentration is >2 mg/mL, the apparent viscosity of the mucin increases in a concentration-dependent manner [100]. Jüngst et al. utilised a Contraves LS-30 coaxial rotational viscometer to assess the viscosity of bile in patients with gallstones. They discovered that the viscosity of gallbladder bile in patients with gallstones was significantly higher than that of liver bile. In addition, the viscosity of gallbladder bile positively correlated with mucin concentration. The physiological concentration of soluble mucin in the gallbladder bile of patients with gallstones and regular people is 0.1–5 mg/mL [101]. In comparison, the concentration of mucin in mucus gel and sludge could be as high as 20 mg/mL [101]. However, when the concentration was >20 mg/mL, the mucin monomers underwent transient gelation, with more non-covalent interactions between the molecules, leading to less reversibility of the gelation behaviour [37]. The high viscosity of the mucus gel and mucin-rich bile can lead to cholestasis, weaken gallbladder activity, delay bile emptying, increase the residence time of stone-causing substances in the gallbladder or bile duct, and promote stone formation (Figure 1e) [102,103].

Cholestasis and impaired gallbladder activity are important factors in gallstone formation (Figure 3b). An increased proportion of MUC5AC and MUC2 in bile can make the mucus in the biliary tract more viscous [94,104]. The multimeric forms of MUC2 and MUC5AC, along with abundant O-linked oligosaccharides, can increase bile viscosity and reduce its flow, thereby accelerating stone formation [26,105]. In addition, excessive mucin in the bile duct can disrupt bile flow and result in the accumulation of bile components, further obstructing the flow of bile and promoting stone growth [106,107].

In clinics, ursodeoxycholic acid (UDCA) is the only well-recognised and widely used drug for the dissolution of gallstones. UDCA is believed to play a role in the treatment of stones by reducing cholesterol absorption in the intestine and secretion of cholesterol in bile [108,109]. In addition, Fischer et al. found that UDCA can reduce the viscosity of gallbladder bile [110], and Kim and Jang found that the combination of n-3 polyunsaturated fatty acids and UCDA reduced the formation of mucin in bile in mice, thereby reducing stone formation [111,112]. The cholagogic effect of UCDA, which increases the flow volume and rate of bile [113], combined with a decrease in mucin expression in the biliary tract, promotes a reduction in existing stones in the biliary tract and inhibits the formation of new stones. Certainly, the effects of these drugs need to be further demonstrated in large-sample randomised controlled trials with long-term follow-ups.

**Table 1 biomolecules-14-00676-t001:** Important studies and results for the role of mucins in gallstone occurrence and development.

First Author’s Name	Publication Year	Study Design	Study Results	Conclusions
Nezam H Afdhal [24]	2004	Fluorescence recovery after photobleaching (FRAP) and fluorescence correlation spectroscopy (FCS) was utilised to examine the role of gallbladder mucin (GBM) in promoting the aggregation and/or fusion of cholesterol-enriched vesicles.	GBM had a profound effect on inducing vesicles to aggregate/fuse.	Both glycosylated and nonglycosylated domains of GBM are involved in early aggregation of cholesterol-enriched vesicles.
P C Sheen [34]	1998	Periodic acid–Schiff–alcian blue double staining was performed for gallbladder specimens to compare the mucin area to the total epithelial area ratio.	Gallbladders with pigment gallstones contained more mucin than gallbladders with or without cholesterol gallstones.	Mucin might play a more important role in the formation of brown pigment stones than that of cholesterol stones.
N H Afdhal [60]	1995	Three kinds of physical and chemical technology were used to quantify the gallbladder mucins affecting vesicle aggregation and fusion.	Without mucin, fusion was slow and took up to 24 h; however, with a physiological concentration of mucin, complete fusion of the vesicles occurred within 6 h.	The mucin hydrophobic domain promotes nucleation of cholesterol monohydrate crystals.
P F Levy [61]	1984	Mucin was added to model bile to observe cholesterol crystal nucleation time.	The number of cholesterol crystals in the 4 mg/mL group was significantly more compared to the other group	Nucleation of hydrated cholesterol crystals in gallbladder bile promotes the formation of cholesterol stones.
Zhitang Lyu [77]	2021	16s rRNA sequencing analysis was performed on patients with gallstones and biliary tract disease and on patients with biliary and duodenal microbial communities.	There were similarities in the abundance of major taxa between the two groups.	The reflux of bacteria in the duodenum into the common bile duct through the sphincter of Oddi should be an important factor in bacterial infection and stone formation in the common bile duct.
Xiaodong, Wu [86]	2021	Different concentrations of LPS stimulation in HIBEpiC and the detection of MUC5AC transcription and secretion level were studied.	LPS upregulated mucin expression in bile duct epithelial cells in a dose-dependent manner.	LPS promotes the secretion of MUC5AC to promote stone formation.
Sven Fischer [110]	2004	The effects of ursodeoxycholic acid (UDCA) on gallbladder bile composition, viscosity, and precipitable fractions were studied in 25 patients with cholesterol gallstones.	The concentrations of protein and mucin in gallbladder bile tended to be lower in UDCA-treated groups.	UDCA treatment reduces total and vesicular cholesterol, the formation of cholesterol crystals, viscosity, and the total amount of sedimentable fractions in gallbladder bile.
Sung Ill Jang [111]	2019	The effect of n-3 polyunsaturated fatty acids (PUFA) in combination with UDCA was evaluated in a mouse model of cholesterol gallstones.	Expression levels of mucin genes were significantly lower in the UDCA, PUFA, and combination groups.	Combination treatment with PUFA and UDCA dissolves cholesterol gallstones in mice by decreasing mucin production, increasing levels of phospholipids and bile acids in bile, and decreasing cholesterol saturation.

## 8. Conclusions

Mucin is involved in the formation and development of cholesterol and pigment stones. Current research suggests that mucin can accelerate the precipitation of cholesterol crystals. Mucins act by forming a high-viscosity gel and mucin network, acting as a matrix and scaffold for stone growth. Bacterial infection and inflammation promote the expression and secretion of mucin to promote the formation and growth of stones further. The high viscosity of mucus gel and mucin-rich bile is also an important factor in promoting gallstone formation. Clarifying the role of mucin in the formation and development of gallstones is helpful for further exploring the related mechanisms and providing new ideas for the intervention of gallstone formation.

## Figures and Tables

**Figure 1 biomolecules-14-00676-f001:**
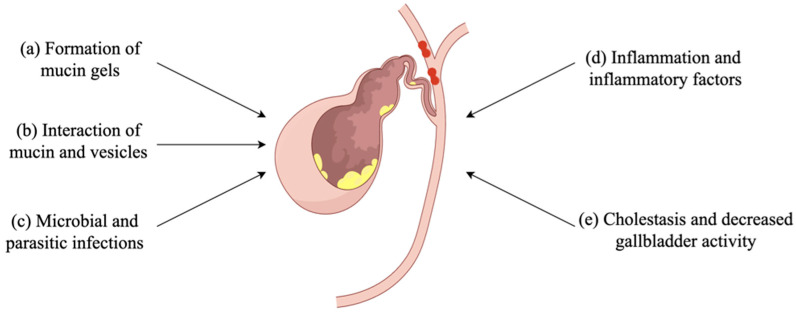
Mucin plays an important role in gallstone formation and development. (**a**) Mucin gel is often considered the precursor of stones, which can be used as a matrix and scaffold for stone growth and participate in stone formation and expansion. (**b**) Mucin can accelerate the fusion and aggregation of vesicles, then forms liposomes and liquid crystals, and finally converts to cholesterol monohydrate crystals. (**c**) Bacterial infection can stimulate excessive secretion of mucin, leading to the formation of stones. (**d**) Inflammation and inflammatory factors increase mucin production in the epithelial cells of the gallbladder and bile duct, contributing to the formation of stones. (**e**) Hypersecretion of mucin causes cholestasis and impaired gallbladder motility, promoting stone expansion. (By Figdraw, https://www.figdraw.com/static/index.html#/, accessed date 18 May 2024).

**Figure 2 biomolecules-14-00676-f002:**
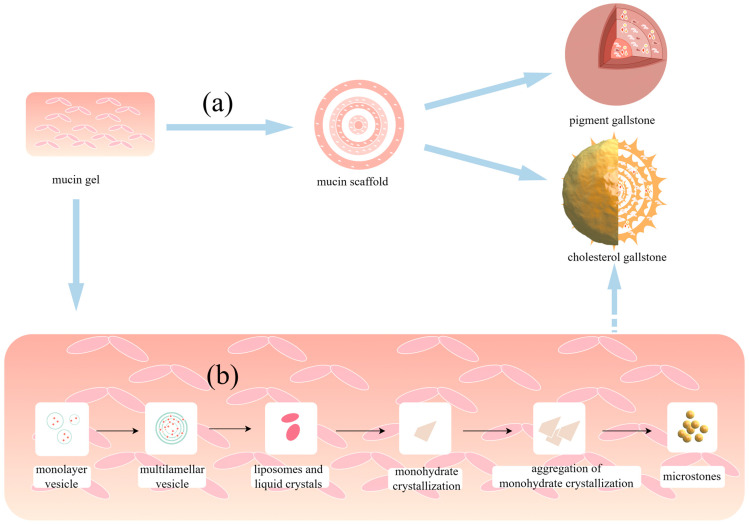
Mechanisms through which mucin contributes to gallstone formation and development: (**a**) Mucins form a highly viscous gel and mucin network, serving as a matrix and scaffold for stone growth and promoting the formation and growth of stones. (**b**) Within the mucin gel, unilamellar vesicles tend to fuse and aggregate, resulting in the formation of multilamellar vesicles that develop into liposomes and cholesterol monohydrate crystals. With the help of mucin gel, cholesterol monohydrate crystals can grow into microstones and eventually into larger macroliths. (By Figdraw, https://www.figdraw.com/static/index.html#/, accessed date includes 18 May 2024).

**Figure 3 biomolecules-14-00676-f003:**
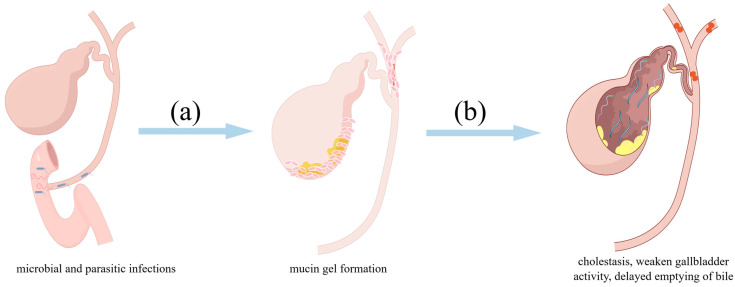
Bacterial infections promote stone formation and growth by increasing mucin secretion. (**a**) Bacterial infections can increase mucin secretion and promote stone formation and growth. (**b**) The high viscosity of mucus gel and mucin-rich bile can lead to cholestasis, weaken gallbladder activity, delay bile emptying, increase the residence time of stone-causing substances in the gallbladder or bile duct, and promote gallstone formation. (By Figdraw, https://www.figdraw.com/static/index.html#/, accessed date 18 May 2024).

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
