# Peer review of "Role of Secretory Mucins in the Occurrence and Development of Cholelithiasis"

_biomolecules, 2024, doi:10.3390/biom14060676_

Round 1

Reviewer 1 Report

Comments and Suggestions for Authors

Topic is of potential clinical interest. Perfect is the aim of the paper, the methods to approach the issues, and the refrence list that is complete and actual. I recommend including some aspects in the revised version.

Minor points: 

1. English and grammar need substantial improvement.

2. Figure 2  is diffuse, I recommend a simple flow chart instead.

3. I am missing clinical approaches  on prophylaxix of gall stone development basen on the results you are discussing. 

4. I am missing comments on factors initiating the formation of gallstones in the intrahepatic bile ducts. 

5. Explain  how LPS enters the bile ducts and the gall bladder. Are serum levels of LPS available? 

6. Are gram-negativ bacteria in the intestinal tract a real problem in patients with cholecystolithiasis? How can this  risk factor be reduced?

7. Please include search terms and sources you used.

Comments on the Quality of English Language

English and grammar need substantial improvement.

Reviewer 2 Report

Comments and Suggestions for Authors

This is a review regarding the role ole of secretory mucins in the occurrence and development

of cholelithiasis. The paper is interesting and clinically relevant. The text is well written, figureas are clear, references are apropriate.

Some issues should be adressed:

1.       The introduction should be shortened and only the most important informations regarding the presented topic should be inserted. The other paragraphs currently included in the introduction, i.e. Mucin is involved in gallstone formation and others should be separate parts following introduction in this manuscript.

2.       The part presenting the physiology of the gallbladder function and bile secretion should be added.

3.       The table summarizing the most important studies described in the test should be added. It should include: first author’s name, publication year, study design, study results and conclusions.

Comments on the Quality of English Language

Minor editing of English language required.

Round 2

Reviewer 2 Report

Comments and Suggestions for Authors

The authors have improved their manuscript according to my suggestions.

Comments on the Quality of English Language

Minor editing of English language required.